# The Use of Medical and Non-Medical Services by Older Inpatients from Emergency vs. Chronic Departments, during the SARS-CoV-2 Pandemic in Poland

**DOI:** 10.3390/healthcare9111547

**Published:** 2021-11-12

**Authors:** Iwona Demczyszak, Justyna Mazurek, Dominik M. Marciniak, Katarzyna Hap, Natalia Kuciel, Karolina Biernat, Edyta Sutkowska

**Affiliations:** 1University Rehabilitation Centre, Wroclaw Medical University, 50-556 Wroclaw, Poland; iwona.demczyszak@umw.edu.pl (I.D.); katarzyna.hap@umw.edu.pl (K.H.); natalia.kuciel@umw.edu.pl (N.K.); edyta.sutkowska@umw.edu.pl (E.S.); 2Department of Drugs Form Technology, Wroclaw Medical University, 50-556 Wroclaw, Poland; dominik.marciniak@umed.wroc.pl

**Keywords:** COVID-19, SARS-CoV-2, public health care, FIMA, elderly, health resources

## Abstract

The COVID-19 pandemic has forced numerous changes in medical care. The monitoring of current needs and problems among the elderly in health care facilities seems to be essential. This study aims to assess the difference in terms of the use of medical and non-medical services before planned or emergency hospitalisation by the older population during the strict lockdown period due to the SARS-CoV-2 pandemic in Poland. The study used the FIMA (Fragebogen zur Inanspruchnahme medizinischer und nicht-medizinischer Versorgungsleistungen im Alter) questionnaire. Patients admitted on a planned basis (*n* = 61) were on average 4 years younger, self-administered the questionnaire more frequently and used the services of different types of therapists. Patients admitted on an emergency basis (*n* = 60) were more likely to visit general practitioners and other specialists and used the carer’s allowance benefits. In the case of the elderly, emergency hospitalisation during the pandemic is more frequently preceded by seeking outpatient care in specialists in various fields, covered by insurance. The chronically ill use the services of various therapists while awaiting hospitalisation, usually not covered by health insurance. For both groups, the age > 73 years is critical for the use of assisting means and completing the self-administered questionnaire, which can be used in planning the health care in these patients.

## 1. Introduction

The turn of March and April 2020 was a period of significant restrictions in Poland due to COVID-19. Ambulation was limited but lockdown also touched medical consultations, rehabilitation, and health resort treatment. The COVID-19 pandemic led to dramatic changes in the provision of routine health care in Poland [1]. The vast part of routine health care was postponed or replaced by online consultations or teleconsultations to prioritise access to hospital beds, staff and ventilators for COVID-19 patients, and to minimise the risk of infection for other patients [2]. Planned hospital admissions were also postponed [1]. This shifted the burden of services from hospitals to outpatient care. In contrast, emergency admissions, which are unpredictable, continued as normal [3]. The pandemic has provided a new context that allowed previously long-held assumptions and standards concerning the manner in which care should be provided to be urgently re-examined and, if necessary, changed [4]. 

Older patients in particular may have felt the changes made during the strict lockdown period as they feared for their health and life. Over 95% of deaths caused by COVID-19 occur in those aged over 60 years. According to WHO, support for the elderly and assessment of their health needs constitute an essential part of the pandemic response [5].

Moynihan et al. reviewed the health system response to the SARS-CoV-2 pandemic in 20 countries. The authors showed that health care utilisation decreased by 37%. The decrease in services overall was made up of reductions in visits (42%), admissions (28%), diagnostics (31%), and therapeutics (30%) [6]. These data suggest that access to diagnostic services and medical advice was significantly limited to prevent the COVID-19 spread and to reduce morbidity and mortality in the outbreak of the pandemic. As older adults are more prone to become infected and are at higher risk of developing serious complications, they have been particularly affected by the new access barriers.

The assessment of health needs is a systematic approach to ensure that the health service uses its resources to improve the health of the population in the most efficient way. The WHO’s definition of health is frequently used: "Health is a state of complete physical, psychological, and social wellbeing and not simply the absence of disease or infirmity” [7]. Health care needs are those that can benefit from health care (health education, disease prevention, diagnosis, treatment, rehabilitation, terminal care) [8]. However, the health care system has been severely affected by the pandemic in many countries, which surprised care givers as well as governments. To manage the health care system reasonably and successfully follow it, it is essential to systematically track the needs of society and relate them to health care opportunities that may be modified by situations such as the COVID-19 pandemic. 

The distinction between individual needs and wider needs of the community is significant for the planning and provision of local health services. If these needs are ignored, there is a danger of a top-down approach to provision of health services, which relies too heavily on what a few people regard as the population needs rather than what they actually are [8]. The purpose of the assessment of health care needs is gathering the information required to bring about changes that are beneficial to the population’s health [9]. Health gain can thus be achieved through the reallocation of resources based on the reassessment of the population’s needs with regard to the changing epidemic situation. 

Although Poland has adopted many preventive strategies and implemented many international recommendations to combat the SARS-CoV-2 pandemic (e.g., preventive measures, maintaining social distancing, early case detection and isolation of patients, testing, epidemic surveillance, protection of high-risk groups, travel control, international cooperation and exchange of experiences public education, a vaccine policy, and many more) [10], there is still much to be improved. Results presented by González-Touya et al. suggest that “Poland, Italy and Greece should pay attention to possible equity problems regarding denied medical care that arose during the first months of the pandemic” [11]. Delays in diagnosis and treatment, especially in elderly patients, may ultimately translate into adverse health outcomes, reduced quality of life, and even widen socioeconomic health inequalities, which will impact a growing share of the European population in the next decades [11].

The subjects of our assessment of health care needs were the populations and patients who are recipients or potential beneficiaries of health care. Populations include individual patients [9]. The Questionnaire for Health-Related Resource Use in an Elderly Population (FIMA, Fragebogen zur Inanspruchnahme medizinischer und nicht-medizinischer Versorgungsleistungen im Alter) [12] is a self-administered questionnaire that assesses nine features and contains 28 questions altogether. Questions 1–18 refer directly to the use of medical and non-medical services by the elderly, whereas questions 19–28 concern the sociodemographic characteristics of the respondent as well as an assessment of the questionnaire difficulty and time spent completing it.

This study aims to assess the differences in terms of the use of medical and non-medical services by the elderly admitted to hospital wards on a planned or emergency basis during the strict lockdown period due to the SARS-CoV-2 pandemic in Poland. Hence, the results of the FIMA questionnaire also reflect the problem of the elderly’s need for these services during lockdown, the period preceding hospitalisation. The specific aim was to assess whether the studied sociodemographic factors (e.g., age, education, or gender of elderly patients) are significantly related to the given areas of the FIMA questionnaire. We also studied if there were any independent factors that influence patient needs.

## 2. Materials and Methods

### 2.1. The Participants and Procedure

The participants were enrolled when hospitalised in one of two departments mentioned below from July to August 2020. The first department was dedicated to chronic diseases and the second functioned as an emergency department, following the conclusion of our agreement (no. 411/2020) with the Bioethics Committee of the Wroclaw Medical University. All of the patients were informed about the aim of the presented study and their rights, and they gave oral permission to participate in the survey. Age > 65 years was the only inclusion criterion. 

Group A consists of patients who were consecutively admitted to the rehabilitation ward at the St. Jadwiga Hospital in Trzebnica, as this type of department always serves for planned hospitalisation. The rehabilitation ward includes 84 beds and is intended for the treatment and rehabilitation of people, mainly the elderly, (1) who have undergone surgery in different areas of the musculoskeletal system (e.g., after joint arthroplasty, fractures, or multi-organ damage) or (2) who suffer from chronic diseases (including, in particular, discopathy, spondylosis, rheumatoid arthritis, or osteoarthritis). The length of stay in the ward is four to six weeks. 

Group B consists of patients who were admitted to the internal medicine and geriatric ward at the A. Falkiewicz Hospital in Wroclaw. The hospital’s ward includes 56 beds and is intended for the treatment of mainly the elderly with acute cardiovascular incidents (stroke, hypertensive crisis, exacerbation of heart failure, etc).

COVID-19-related hospitalisation was the main exclusion criterion. The other exclusion criteria included severe eye or ear diseases, a life-threatening condition, cognitive impairment preventing the patient from understanding the FIMA questionnaire (assessed by the Mini-Mental State Examination, MMSE, <26 points), and refusal to participate in the study. 

All of the patients were asked to participate in the study and fill in the FIMA questionnaire within 24 h after admission. We used the areas of the FIMA questionnaire that assess the use of medical and non-medical services within the past 3 months and those that do not impose a time frame to assess the use of these services by the elderly of the two study groups during the strict lockdown period in Poland (possible change in the use of these services). The authors omitted the analysis of the areas of the FIMA questionnaire where the patient is asked to rate their use of medical and non-medical services over the past 12 months because this period did not cover the lockdown period. 

### 2.2. Statistical Analysis

The description of the analysed variables in the ratio scales (age) was performed by calculating basic descriptive statistics for them: mean value, variance, standard deviation, standard error, skewness, median, range, lower and upper quartiles, 95% confidence interval (95% ± CI) for mean and standard deviation. The normality of the distributions of these variables was assessed using the Shapiro–Wilk test, analysing both raw and logarithmically transformed data. The primary statistical inference was based on the results of two different computational procedures that are appropriate for the analysis of dependent variables based on dichotomous scales: (1) the nonparametric analysis using Pearson’s chi-squared statistics calculated for multivariate tables and (2) the univariate logistic regression. When the expected values, which were calculated based on the chi-squared statistics, were less than 10, the Yates correction for small numbers was considered when assessing the statistical significance of differences between the compared subgroups. Moreover, the statistical significance of interactions between independent variables, whose values could be represented on a dichotomous scale, was assessed using the chi-squared statistics-based meta-analysis that was calculated according to the random effects model. For each of the dichotomous independent variables considered in the developed random-effects meta-analysis model, the odds ratio (OR), its statistical significance (p), 95% confidence interval, and the percentage contribution of each dichotomous independent variable to explain the variation in the model’s variance were determined. A parametric model of the multivariate analysis of variance, the ANOVA/MANOVA model, was used for assessing the effect of belonging to one of groups, A or B, and the type of specialist visited (FIMA question 1). The least significant difference (LSD) test was used for multiple post hoc comparisons. The normality of the distributions of the compared variables was determined using the Shapiro–Wilk test, and the homogeneity of their variances using the Levene’s test. 

A significance level of *p* = 0.05 was adopted in all statistical analyses performed. Statistical analyses were performed using the STATISTICA PL ® version 13 software (StatSoft Polska Sp. z o.o., Cracow, Poland).

## 3. Results

In the defined period, 65 patients attended the hospital (Trzebnica) for planned hospitalisation and 66 patients attended the hospital in Wroclaw for emergency reasons. In the Trzebnica’s ward, four patients withdrew their consent during the selection period. In the Wroclaw’s ward, two patients declined to fill in the FIMA questionnaire after they had read its content and four patients were excluded due to MMSE < 26 pts. Therefore, the final analysis of health needs according to FIMA was performed on 61 patients (women, *n* = 51, 84%; men, *n* = 10, 16%) from the hospital in Trzebnica (planned hospitalisation) and 60 patients (women, *n* = 40, 66.67%; men, *n* = 20, 33.33%) from the hospital in Wroclaw (emergency admission). 

### 3.1. Socio-Demographic Data

The preliminary comparative analysis of the studied groups A and B revealed their homogeneity (*p* > 0.05) in terms of three demographic data: sex, marital status, and education. Women prevailed in both groups. The majority of the patients in both groups declared secondary education level (A, 47.54%; B, 40%). The analysis of the marital status revealed that 63.93% patients in group A and 50% in group B were in a steady relationship. The data are shown in Table 1 and Table 2. 

The performed statistical analysis using the parametric Student’s *t*-test found a statistically significant difference between the mean age values in the compared groups (*p* = 0.00097). The mean age of patients in group A was 4 years lower than that in group B. The mean ages were 71.74 ± 5.7 years for group A and 75.87 ± 7.6 years for group B, respectively, and the age ranges were 65–87 and 65–94 years.

The statistical analysis using the chi-squared test revealed that patients in group A more frequently declared the ability to complete the self-administered FIMA questionnaire (75.41%, *n* = 46) than patients in group B (55%, *n* = 33 completed the self-administered FIMA questionnaire). The limiting factor for completion of the self-administered FIMA questionnaire in both groups was age, which was set at 73 years (Table 3). The Receiver Operating Characteristic (ROC) curve analysis was used for determination of this cut-off, and the cut-off point was calculated by minimising the error rate using Youden’s method.

Age 73 was also found to be critical for the likelihood of using assisting means (orthopaedic aids and supplies; FIMA questionnaire differentiates them into walker, wheelchair, stair lift, bath lift, glasses/vision aid, hearing aid, dentures, oxygen unit, sleep apnea treatment unit, compression stockings, regular use of incontinence pads, and other). The meta-analysis revealed that this likelihood is on average 1.63 times higher in the age group of people older than 73 years compared to younger people. Wheelchair use (OR = 10.2; *p* = 0.03) and hearing aid use (OR = 12.0; *p* = 0.02) are most likely (Figure 1).

Furthermore, for those who had difficulty completing the FIMA questionnaire on their own, the likelihood of using assisting means was 2.5 times higher than in the group who had no difficulty in completing the self-administered survey. High OR values in subgroups such as wheelchair, bath seat, and hearing aid imply that these are motor rather than intellectual problems. The effect of such defined age on used assisting means is shown in Figure 2.

### 3.2. FIMA

The statistical analysis using the non-parametric chi-squared test, the results of which are shown in Table 4, found that the compared groups A and B differ in terms of their answers to FIMA 1, FIMA 2, and FIMA 8 questions.

The analysis showed that belonging to group A or B correlates with the type of specialty of a visited physician, which was the focus of FIMA question 1. Patients in group B were relatively more likely to visit both GPs (General Practitioner) and other specialists. However, they most frequently visited the GP’s office (Table 4, Figure 3).

Patients in group A were more likely to have used the services of therapists such as a physiotherapist, ergotherapist or occupational therapist, speech therapist, podiatrist, or osteopath/unconventional medicine/acupuncture within the past 3 months prior to hospitalisation, as indicated by the FIMA question 2 (Table 4).

Patients in group B were more likely to report using their carer’s allowance benefits, e.g., attendance allowance, attendance benefit (FIMA question 8) (Table 4).

## 4. Discussion

The SARS-CoV-2 coronavirus pandemic, which has been ongoing for many months, is severely impacting the health of the elderly in many dimensions [13]. COVID-19 changes daily routines and significantly reduces the quality of life in the elderly [14]. The epidemiological situation was creating new barriers to accessing health services day by day. 

This study, using selected areas of FIMA, analysed health needs and differences in terms of services provided to the elderly hospitalised during the Sars-CoV-2 pandemic in Poland. The assessment covered a period of 3 months of strict lockdown in Poland, which preceded hospitalisation of the surveyed patients who were finally admitted to hospital on a planned or emergency basis. This allowed us to investigate how the elderly find themselves during a time of significant reduction in access to medical services according to the type of condition. 

Patients admitted on an emergency basis revealed a significantly lesser ability to complete the self-administered FIMA questionnaire compared to patients admitted on a planned basis. The results of this observation are reflected in the literature, where many authors emphasise that acute illness in the elderly is very often associated with greater dependence on basic activities of daily living [15,16,17]. Hospitalisation of the elderly due to disease exacerbations or life-threatening conditions results in decreased motor and cognitive function, especially impaired memory processes, attention processes, information processing speed, and performance of target tasks [18]. The results presented in this study suggest an absolute need to make an early prediction of loss of independence during and after emergency hospitalisation. 

As we age, the condition of our body changes. As we enter old age, functions of our body are significantly weakened. A meta-analysis of all surveyed patients was conducted to search for an age that was independent of the type of hospitalisation and type of condition (acute or chronic). The meta-analysis revealed that patients older than 73 years were on average 1.63 times more likely to need assisting means than younger patients. These results are in line with the WHO’s 2020 Global Report [19], according to which older adults aged 70 years and older have twice the rate of loss of functional ability and mental capacity. Every older adult should have the opportunity of early optimisation of functional ability and mental capacity to eliminate health-related problems, and the critical point appears to be the age of 73 years. It is necessary to implement appropriate action programmes on healthy aging and health care for the elderly’s diverse needs. Needs assessment is the fundamental base for an effective social and health policy and the provision of needs-based interventions. Unmet needs can lead to a decreased quality of life and increased costs of care. It is important to be aware of the subjective needs of the elderly people and to meet them to improve their quality of life and to provide more appropriate person-centred care and support.

Furthermore, studies show that the estimated percentage of severe and moderate loss of functional ability and mental capacity is higher in older women than in older men [19]. There is a need for further research concerning the difference resulting from the ageing of the body in terms of sex. 

A detailed analysis of the results of the FIMA questionnaire revealed a difference in terms of the use of medical visits between groups during the pandemic period. Chronic patients did not use medical visits at all during this period, while emergency patients reported visiting both GPs and specialists. It should be noted, however, that the group of chronic patients was significantly younger than the group of emergency patients. Therefore, it cannot be excluded that less frequent outpatient visits were caused by a lower prevalence rate, which is associated with younger age. As medical consultations changed their form in the pandemic era (teleconsultations, video consultations) [20], younger patients may also have been better able to manage the technical difficulties of such consultations, which the FIMA questionnaire does not examine. However, the differences reported in this study also may indicate other problems that were observed in Poland even before the period of Sars-Cov-2 virus spread. In Poland, access to medical services varies widely [21]. The Supreme Chamber of Control (Polish: NIK) and expert analyses in a 2014 report list many factors that affect access to medical services: place of residence, age, material status, level of education, and uneven distribution of the system infrastructure (structure of services provided) [21]. The patient groups in this study vary in age and place of health service delivery. The studied group of emergency patients was treated in a hospital in Wroclaw (a provincial city with a population of nearly 700,000 inhabitants [22]), while chronic patients were treated in a hospital in Trzebnica (a city in the Lower Silesia Province with a population of nearly 14,000 inhabitants [22]). The likelihood of accessing health care in a large provincial city is undoubtedly higher than in a smaller city. There are areas in Poland where accessibility to health services is low, and there are regions that offer greater access to health services [21,23]. 

Certainly, the higher number of consultations in the group of emergency patients may also be due to the fact that these individuals actively sought (despite restrictions) outpatient care for their unmet therapeutic needs. In contrast, chronic patients were likely to be able to manage their condition and they were able to postpone their visits to either their GP or a specialist for fear of COVID-19 infection. It cannot be ruled out that in a non-pandemic period, patients in group B would be hospitalised more quickly (either by the referring doctor or by the patient themselves) and that the number of consultations in the period before hospitalisation would be lower. This thesis is supported by our study that compared the use of medical visits during and before the lockdown period among a representative group of chronically ill patients in the same hospital in Trzebnica [20].

Another observed variable was the use of services of one of listed therapists (physiotherapist, ergotherapist or occupational therapist, speech therapist, podiatrist, osteopath/unconventional medicine/acupuncture) within the past 3 months. When comparing the studied groups of the elderly, it was observed that almost triple the studied patients (9.92%) treated on a planned basis declared visiting one of the listed therapists compared to emergency patients (3.31%). These results can be certainly explained by the type of ward in Trzebnica, where studied patients were hospitalised. It is a rehabilitation-oriented unit and its patients often receive outpatient therapy for primarily musculoskeletal conditions while awaiting inpatient stays. 

There was also a statistically significant difference between the studied groups in terms of the use of carer’s allowance benefits, e.g., attendance allowance and attendance benefit. Emergency patients were more likely to declare using such assistance compared to chronic patients (13.23% vs. 4.13%). Most likely, the age difference between the studied groups also had a significant effect on the obtained results (worse independence and higher prevalence rate). The progressive aging of the population determines the need for more and more people to receive their carer’s allowance and attendance allowance that should be provided during the long-term care. 

Further observation and analysis of the differences in terms of the access to medical and non-medical services by the elderly is necessary not only for the ongoing correction of the tasks of senior policy, but also for the discussion of its post-pandemic shape.

As there are differences in terms of the use of social insurance benefits and commercial benefits across countries, and perhaps in terms of even age groups or medical conditions, there is a need to consider these differences in further research. 

Most researchers in the field of gerontology agree that geriatric medicine should be developed with the promotion of specialists in the field and an increase in the number of departments and beds dedicated to the care of older patients. Systemic and institutional actions must be directed towards the promotion, prevention, and health protection of seniors. The primary subsidisation of social welfare may in the near future result in the generation of savings due to a lack of necessity to cover some persons with costly services provided in 24-h care facilities or with frequently repeated medical hospitalisations. The network of “support and activation institutions” for the elderly should be further expanded, such as with seniors’ clubs and day care centres, day care centres for seniors, and support for services possibly provided at the home of an elderly person. Especially in this pandemic era and beyond, it is important to pay attention to the mental health of seniors. A digital education program for older people would enable them to acquire new digital competences that could reduce the negative effects of isolation and loneliness. “Older people online” can sustain social relationships that are increasingly made online and significantly improve safety in emergency situations [21,24,25,26].

## 5. Conclusions

In the case of the elderly, emergency hospitalisation during the pandemic is more likely to be preceded by seeking outpatient care among specialists in various disciplines. These individuals primarily use services that are covered by insurance. Despite the pandemic, the chronically ill use the services of various therapists while awaiting hospitalisation, which are frequently not covered by health insurance. The age of 73 years is critical for the use of assisting means, as is the completion of the self-administered questionnaire designed for the elderly. This information can be used in planning the prevention of a lack of independence in these patients despite the COVID-19 situation. 

### Limitations: 

Small size of study groups.No reference of the same groups to the pre-lockdown period.The FIMA questionnaire is used for assessing the use of various areas of medical and non-medical services; however, it does not analyse the reasons for the collected observation data.

## Figures and Tables

**Figure 1 healthcare-09-01547-f001:**
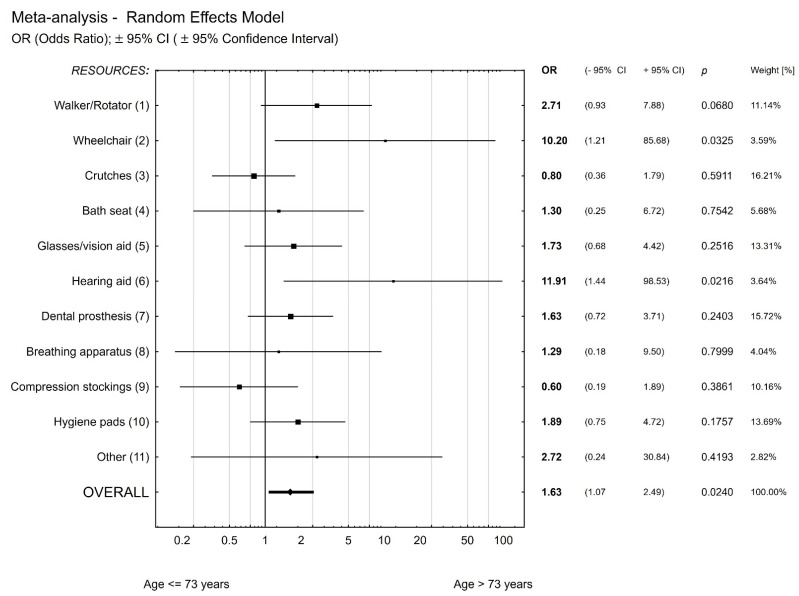
Likelihood of the use of assisting means in people 73+.

**Figure 2 healthcare-09-01547-f002:**
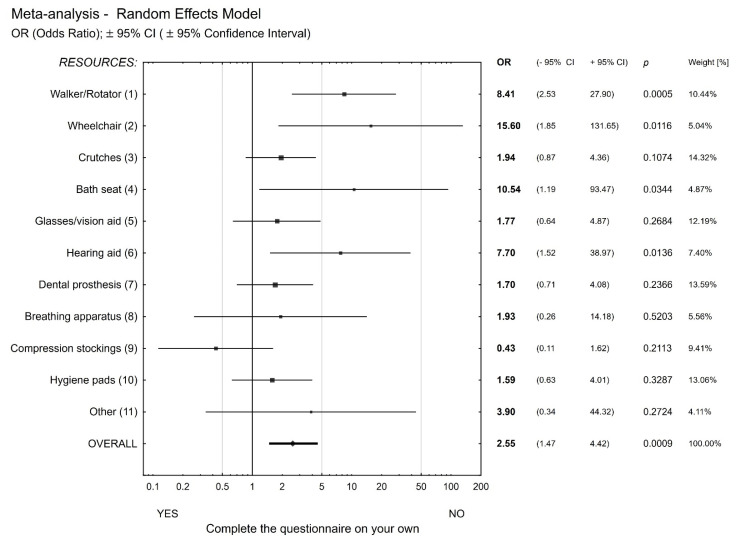
The use of assisting means versus the ability to complete the self-administered survey.

**Figure 3 healthcare-09-01547-f003:**
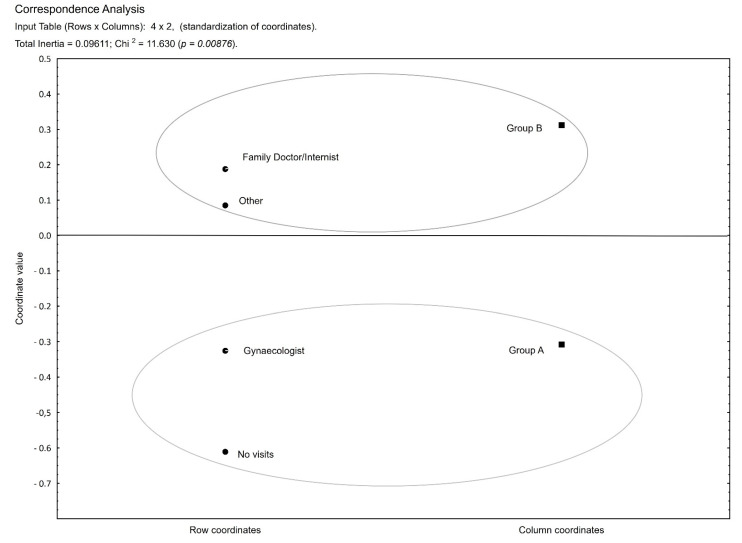
Correlation: group membership versus a type of specialty of a visited physician.

**Table 1 healthcare-09-01547-t001:** Baseline characteristics for both groups.

Dependent Variable:		Type of Group				Pearson’s Chi^2^	*p*-Value	OR
		Group A		Group B				
Sex (F = 1/M = 2)	Subgroup:	F	M	F	M	3.72	0.0539	2.27
	N	50	11	40	20			
Marital status (in relationship: YES = 1/NO = 0)	Subgroup:	0	1	0	1	2.40	0.1216	0.56
	N	22	39	30	30			

Abbreviations: F, female; M, male; OR, odds ratio; N, number of individuals.

**Table 2 healthcare-09-01547-t002:** Baseline characteristics for both groups.

Dependent Variable:		Type of Group								Pearson’s Chi^2^	*p-*Value
		Group A				Group B					
Education	Subgroup:	Secondary	Primary	Vocational	Higher	Secondary	Primary	Vocational	Higher	5.27	0.1530
	N	29	8	17	7	24	13	10	13		

Abbreviations: N, number of individuals.

**Table 3 healthcare-09-01547-t003:** Size in both groups according to age distribution.

Age	Type of Group Group A N	Type of Group Group B N	Line Total N
Age > 73	17	36	53
Age ≤ 73	44	24	68
Total	61	60	121

Abbreviations: N, number of individuals.

**Table 4 healthcare-09-01547-t004:** Selected FIMA areas concerning the use of medical and non-medical services within the past 3 months.

Lp	Dependent Variable:		Type of Group				Pearson’s Chi^2^	*p*-Value	OR
			Group A		Group B				
	Subgroup:		0	1	0	1			
1	FIMA 1 (NO = 0/YES = 1) Have you visited any of the following physicians within the last 3 months?	N	17	44	4	56	9.4800	*p* = 0.00208	5.40909
		%	14.05%	36.36%	3.31%	46.28%			
*2*	FIMA 2 (NO = 0/YES = 1) Have you used services of any of the following therapists within the last 3 months?	N	49	12	56	4	4.4587	*p* = 0.03472	0.29167
		*%*	40.50%	9.92%	46.28%	3.31%			
3	FIMA 3 (NO = 0/YES = 1) Have you used home services of a community nurse or social worker within the last 3 months?	N	55	6	54	6	0.0009	*p* = 0.975940	1.01852
		%	45.45%	4.96%	44.63%	4.96%			
4	FIMA 4 (NO = 0/YES = 1) Have you used paid caregiving services within the last 3 months due to your health status?	N	59	2	59	1	0.33	*p* = 0.56856	0.50000
		*%*	48.76%	1.65%	48.76%	0.83%			
5	FIMA 5 (NO = 0/YES = 1) Have you used informal assistance within the last 3 months due to your health status?	N	34	27	27	33	1.40	*p* = 0.23754	1.53910
		%	28.10%	22.31%	22.31%	27.27%			
6	FIMA 8 (NO = 0/YES = 1) Do you use the carer’s allowance benefits?	N	56	5	44	16	7.19	*p* = 0.00731	4.07273
		%	46.28%	4.13%	36.36%	13.22%			
7	FIMA 9 (NO = 0/YES = 1) Have you taken any medications within the last 7 days?	N	3	58	0	60	3.03	*p* = 0.08195	“-”
		%	2.48%	47.93%	0.00%	49.59%			
		*%*	32.23%	18.18%	47.11%	2.48%			
8	FIMA 14 (NO = 0/YES = 1) Do you have one or more assisting measures?	N	3	58	38	22	0.00	*p* = 0.98343	0.98276

Abbreviations: OR, odds ratio; N, number of individuals.

## Data Availability

The data presented in this study are openly available at https://pl.padlet.com/edytasutkowska/r1665dcjtisz0kz2 (Accessed 12 October 2021).

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
