# Peer review of "The Use of Medical and Non-Medical Services by Older Inpatients from Emergency vs. Chronic Departments, during the SARS-CoV-2 Pandemic in Poland"

_healthcare, 2021, doi:10.3390/healthcare9111547_

Round 1

Reviewer 1 Report

Thank you for the opportunity to review this timely and relevant study.  The global challenge of population aging (Jakovljevic et al., 2021) validates research on the elderly in health care facilities to optimize both health care delivery (Li et al., 2021; van Vuuren et al., 2021; Wernly et al., 2021) and public health policy (Heckman et al., 2021; Nelson, 2021; Serikbayeva et al., 2021).  As the authors conclude:  "Further observation and analysis of the differences in terms of the access to medical and non-medical services by the elderly is necessary not only for the ongoing correction of the tasks of senior policy, but also for the discussion of its post pandemic shape."  

Opportunities for improvement include:

  • Clarify definition and nuances of "assisting means"
  • Elaborate on "appropriate action programmes on healthy aging and health care for the elderly's diverse needs."

References:

Heckman, G. A., Saari, M., McArthur, C., Wellens, N. I., & Hirdes, J. P. (2021). RE: COVID-19 response and chronic disease management.

Jakovljevic, M., Westerman, R., Sharma, T., & Lamnisos, D. (2021). Aging and Global Health. Handbook of Global Health, 73-102.

Li, C. M., Lin, C. H., Li, C. I., Liu, C. S., Lin, W. Y., Li, T. C., & Lin, C. C. (2021). Frailty status changes are associated with healthcare utilization and subsequent mortality in the elderly population. BMC Public Health21(1), 1-12.

Nelson, M. A. (2021). The timing and aggressiveness of early government response to COVID-19: Political systems, societal culture, and more. World Development146, 105550.

Serikbayeva, B., Abdulla, K., & Oskenbayev, Y. (2021). State capacity in responding to COVID-19. International Journal of Public Administration44(11-12), 920-930.

van Vuuren, J., Thomas, B., Agarwal, G., MacDermott, S., Kinsman, L., O’Meara, P., & Spelten, E. (2021). Reshaping healthcare delivery for elderly patients: the role of community paramedicine; a systematic review. BMC Health Services Research21(1), 1-15.

Wernly, B., Beil, M., Bruno, R. R., Binnebössel, S., Kelm, M., Sigal, S., ... & Jung, C. (2021). Provision of critical care for the elderly in Europe: a retrospective comparison of national healthcare frameworks in intensive care units. BMJ Open11(6), e046909.

Author Response

REPLY TO COMMENTS OF THE REVIEWER 1

Reviewer #1:

We are grateful for the positive and constructive comments that originated in the review process. We have carefully reviewed the comments and have revised the manuscript accordingly.

 Our  responses are given in a point-by-point manner below. Changes to the manuscript are marked red.

Thank you for the opportunity to review this timely and relevant study.  The global challenge of population aging (Jakovljevic et al., 2021) validates research on the elderly in health care facilities to optimize both health care delivery (Li et al., 2021; van Vuuren et al., 2021; Wernly et al., 2021) and public health policy (Heckman et al., 2021; Nelson, 2021; Serikbayeva et al., 2021).  As the authors conclude:  "Further observation and analysis of the differences in terms of the access to medical and non-medical services by the elderly is necessary not only for the ongoing correction of the tasks of senior policy, but also for the discussion of its post pandemic shape."  

Opportunities for improvement include:

  • Clarify definition and nuances of "assisting means"

RESPONSE: Thank you for this suggestion. We have clarified the definition of “assisting means”: orthopedic aids and supplies (FIMA Questionnaire differentiates them into: Walker, Wheelchair, Stair lift, Bath lift, Glasses/vision aid, Hearing aid, Dentures, Oxygen unit, Sleep apnea treatment unit, Compression stockings, Regular use of incontinence, Pads and Other.

  • Elaborate on "appropriate action programmes on healthy aging and health care for the elderly's diverse needs."

RESPONSE: We are grateful for your suggestion. Of course we have elaborated it in the Manuscript as follows:

It is necessary to implement appropriate action programmes on healthy aging and health care for the elderly's diverse needs. Needs assessment is the fundamental base for an effective social and health policy and the provision of needs-based interventions. Unmet needs can lead to a decreased quality of life and increased costs of care. It is important to be aware of the subjective needs of the elderly people and to meet them to improve their quality of life and to provide more appropriate person-centered care and support.

Reviewer 2 Report

This is a descriptive article based on a single questionnaire and two groups that encompass two somewhat opposing processes, chronicity and urgent illness. This is a key factor as there will be important differences in health care needs.

The topic is interesting but falls short in the sense of what to do with these data, a proposal for a re-evaluation of the needs with the data obtained would be interesting.

In the discussion section, there is a lack of studies of this type to improve the quality of care and the optimization of healthcare resources.
This section goes on to explain the results obtained in the study and the comparison between groups. It would be interesting to provide some more references.

Author Response

REPLY TO COMMENTS OF THE REVIEWER 2

Reviewer #2:

We are grateful for the positive and constructive comments that originated in the review process. We have carefully reviewed the comments and have revised the manuscript accordingly. A revision of the paper has been carried out to take all of them into account, and in the process, we believe the paper has been significantly improved.

 Our responses are given in a point-by-point manner below. Changes to the manuscript are marked red.

This is a descriptive article based on a single questionnaire and two groups that encompass two somewhat opposing processes, chronicity and urgent illness. This is a key factor as there will be important differences in health care needs.

RESPONSE: Thank you very much for pointing this out. We agree that the patients described in the groups differ in terms of disease entities, which may be associated with different needs. But these needs are not known up to now that is why we would like to detect them, mainly in the context of COVID-19 what was the new situation. Please find that “to know” is not enough as we have to prove our expectations. The questionnaire gives us this opportunity.  The authors of the study were guided by a common denominator, it is a fact that the respondents were elderly and a questionnaire was created for them in order to analyze and learn about their healthcare needs in various situations and with various ailments. This knowledge will enable a broader view and appropriate meeting the needs of the elderly, depending on the condition.

The topic is interesting but falls short in the sense of what to do with these data, a proposal for a re-evaluation of the needs with the data obtained would be interesting.

RESPONSE: Thank you very much for this suggestion. We believe that the results of our study confirm the shortcomings in meeting the needs of the elderly in Poland. And, of course, we agree with the need for further analysis of data that could influence further senior policy, we added this issue to the discussion. The authors have plans and needs for further research and interpretation of the questionnaire.

In the discussion section, there is a lack of studies of this type to improve the quality of care and the optimization of healthcare resources. This section goes on to explain the results obtained in the study and the comparison between groups. It would be interesting to provide some more references.

RESPONSE:  Thank you very much for this suggestion. We added it to the discussion section and supplemented with literature.

Reviewer 3 Report

  1. Specific objectives are not clearly indicated.
  2. There is not enough supporting information or evidence regarding what is known/not known in the Introduction section that justifies what this work adds.
  3. There is no proper timeline of activities and interventions against the COVID-19 outbreak in Poland.
  4. Trzebnica and Wrocław are two different cities in Poland. The analysis of health needs according to FIMA was performed on 61 patients from the hospital in Trzebnica (planned hospitalization) and 60 patients from the hospital in Wrocław (emergency admission). These two cities can differ from each other and could have different healthcare and epidemiological situations; thus, it would be necessary to clarify the selection of these two cities and whether there are social, epidemiological similarities, or differences between these two cities.
  5. A more complex problem is to clarify the decision to compare planned hospitalization and emergency admission from two different hospitals, as patients requiring emergency care are different from individuals having chronic diseases.
  6. In fact, and according to the article: “Patients admitted on an emergency basis revealed significantly less able to complete the self-administered FIMA questionnaire compared to patients admitted on a planned basis”. Overall, emergency patients have more severe conditions compared to patients admitted on a planned basis. Thus, probably it is not right to compare these two groups, as emergency patients have fewer chances to complete the questioner because of the severity of the condition compared to another group.
  7. The authors do not justify the novelty in this study and what is adding to the existing literature. Overall results are quite obvious. For instance: “Certainly, the higher number of consultations in the group of emergency patients may also be due to the fact that these individuals actively sought (despite restrictions) outpatient care for their unmet therapeutic need. In contrast, chronic patients were likely to be able to manage their condition and they were able to postpone their visits to either their GP or a specialist for fear of COVID-19 infection”.
  8. The conclusion and recommendations are vague and do not address specific problems.
  9. No citation for many statements.

Author Response

REPLY TO COMMENTS OF THE REVIEWER 3

Reviewer #3:

We are grateful for the positive and constructive comments that originated in the review process. We have carefully reviewed the comments and have revised the manuscript accordingly. A major revision of the paper has been carried out to take all of them into account, and in the process, we believe the paper has been significantly improved.

 Our responses are given in a point-by-point manner below. Changes to the manuscript are marked red.

  1. Specific objectives are not clearly indicated.

RESPONSE: Thank you for pointing this out. We agree with the Reviewer and have added specific aim of our study:

The specific aim was to assess whether the studied sociodemographic factors (e.g. age, education or gender of elderly patients) are significantly related to the given areas of the FIMA questionnaire. We would like also study is there any independent factor/s which influence patient’s needs

  1. There is not enough supporting information or evidence regarding what is known/not known in the Introduction section that justifies what this work adds.

RESPONSE: Thank you for paying attention to this important fact. The section: introduction has been modified and corrected in line with the Reviewer's comments.

  1. There is no proper timeline of activities and interventions against the COVID-19 outbreak in Poland.

RESPONSE: Information about interventions against the COVID-19 outbreak in Poland has been added to the section: introduction.

  1. Trzebnica and Wrocław are two different cities in Poland. The analysis of health needs according to FIMA was performed on 61 patients from the hospital in Trzebnica (planned hospitalization) and 60 patients from the hospital in Wrocław (emergency admission). These two cities can differ from each other and could have different healthcare and epidemiological situations; thus, it would be necessary to clarify the selection of these two cities and whether there are social, epidemiological similarities, or differences between these two cities.

RESPONSE: We would like to stress that patients in Trzebnica and WrocÅ‚aw have the same health problems and diseases. The principles of granting emergency and planned health services are provided on the same terms, regardless of the region/province. This is because patients in Poland are covered by the same insurance. Additionally, both mentioned cities are located in the same province  and are near to each other (about 20 km).

  1. A more complex problem is to clarify the decision to compare planned hospitalization and emergency admission from two different hospitals, as patients requiring emergency care are different from individuals having chronic diseases.

RESPONSE: During the COVID-19 period, the department in Trzebnica had only planned admissions, the ward in WrocÅ‚aw was only dedicated to the  emergency admissions. Moreover, please find that  the surveyed patients were similar in terms of sociodemographic data. Please also see the answer for this question above ( the answer for question number 4).

  1. In fact, and according to the article: “Patients admitted on an emergency basis revealed significantly less able to complete the self-administered FIMA questionnaire compared to patients admitted on a planned basis”. Overall, emergency patients have more severe conditions compared to patients admitted on a planned basis. Thus, probably it is not right to compare these two groups, as emergency patients have fewer chances to complete the questioner because of the severity of the condition compared to another group.

RESPONSE: The FIMA questionnaire is a simple tool that can be used among the elderly people, regardless of the type of hospitalization, with the exclusion of severe cognitive impairment, as suggested by original FIMA author - Dr. Seidl. The aim of our research was to identify the differences in unmet areas in the use of medical and non-medical services by the elderly people admitted to hospitals in two different ways, to indicate also possible gaps in the health care system in Poland and the need for further, in-depth research in this area on a much larger scale on a much larger group of respondents.

  1. The authors do not justify the novelty in this study and what is adding to the existing literature. Overall results are quite obvious. For instance: “Certainly, the higher number of consultations in the group of emergency patients may also be due to the fact that these individuals actively sought (despite restrictions) outpatient care for their unmet therapeutic need. In contrast, chronic patients were likely to be able to manage their condition and they were able to postpone their visits to either their GP or a specialist for fear of COVID-19 infection”.

RESPONSE: Thank you for this suggestion. We agree with that. We have tried to justify the novelty of our study in the section: introduction and have added more literature. We hope, this is  sufficient to show the relevance of our research. We are also aware of the limitations of our study as noted in the manuscript.

  1. The conclusion and recommendations are vague and do not address specific problems.

RESPONSE: Our study was dedicated to the specific situation which COVID-19 is. It was also dedicated to the specific age group, which seems to be the most problematic in the context in dealing with  lockdown. Thus the study addressed  the specific problem: eldery people with chronic or acute diseases and lockdown restrictions. We are convinced that we need a lot of new information about Covid-19 and different groups of patients for better planning the healthcare and social care in the future. We find it particularly valuable to obtain information from our research about the age at which most patients require additional assisting means. This is an observation that is independent of the lockdown situation and may be useful in an overall geriatric assessment when confirmed in a larger group of people. We underline this in the conclusions section.

  1. No citation for many statements.

RESPONSE: Thank you for this important suggestion. The latest references have been added to the sections: Introduction and Discussion.

Round 2

Reviewer 3 Report

The new version of the manuscript has improved significantly.